# Extraction of High-Quality RNA from *S. aureus* Internalized by Endothelial Cells

**DOI:** 10.3390/microorganisms11041020

**Published:** 2023-04-13

**Authors:** Michelle Maurer, Tilman E. Klassert, Bettina Löffler, Hortense Slevogt, Lorena Tuchscherr

**Affiliations:** 1Institute for Medical Microbiology, Jena University Hospital, 07747 Jena, Germany; 2Department of Respiratory Medicine, Medizinische Hochschule Hannover, 30625 Hannover, Germany; tilman.klassert@helmholtz-hzi.de (T.E.K.); slevogt.hortense@mh-hannover.de (H.S.); 3Helmholtz Centre for Infection Research, 38124 Braunschweig, Germany

**Keywords:** intracellular *S. aureus*, host–pathogen interaction, RNA extraction, real-time PCR

## Abstract

*Staphylococcus aureus* evades antibiotic therapy and antimicrobial defenses by entering human host cells. Bacterial transcriptomic analysis represents an invaluable tool to unravel the complex interplay between host and pathogen. Therefore, the extraction of high-quality RNA from intracellular *S. aureus* lays the foundation to acquire meaningful gene expression data. In this study, we present a novel and straightforward strategy to isolate RNA from internalized *S. aureus* after 90 min, 24 h, and 48 h postinfection. Real-time PCR data were obtained for the target genes *agrA* and *fnba*, which play major roles during infection. The commonly used reference genes *gyrB*, *aroE*, *tmRNA*, *gmk*, and *hu* were analyzed under different conditions: bacteria from culture (condition I), intracellular bacteria (condition II), and across both conditions I and II. The most stable reference genes were used for the normalization of *agrA* and *fnbA*. Delta C_q_ (quantification cycle) values had a relatively low variability and thus demonstrated the high quality of the extracted RNA from intracellular *S. aureus* during the early phase of infection. The established protocol allows the extraction and purification of intracellular staphylococcal RNA while minimizing the amount of host RNA in the sample. This approach can leverage reproducible gene expression data to study host–pathogen interactions.

## 1. Introduction

*Staphylococcus aureus* is one of the leading causes of acute and chronic infections worldwide [1]. *S. aureus* can evade antimicrobial defenses by infecting human host cells and establishing an intracellular lifestyle in a variety of cell types including nonprofessional phagocytic cells such as epithelial cells and endothelial cells [2,3]. Within host cells, *S. aureus* is able to switch to a dormant bacterial phenotype called small colony variants (SCVs). This bacterial phenotype is less sensitive to antimicrobial treatment and is related to *S. aureus*’s persistence [4,5,6]. Transcriptomic analyses have proven to be invaluable to unravel the complex interplay of pathogens and eukaryotic hosts [7,8]. Investigating gene expression and concomitant physiological changes of both pathogens and host cells provides insights into (i) how *S. aureus* adapts to intracellular survival and (ii) how the host cell reacts to bacterial invasion. However, accessing bacterial mRNA for transcriptome analysis is a complicated procedure for all researchers. There are many reasons why the isolation of bacterial RNA in general has been technically challenging in the past and present. For example, bacterial mRNA is more prone to degradation than eukaryotic mRNA and has a shorter half-life [9]. In addition, Gram-positive bacteria such as *S. aureus* are bounded by a thick cell wall and, as a result, require mechanical or proteolytic disruption to access the genetic material. Stead and colleagues have developed a method called RNAsnap^TM^, which allows for the inexpensive recovery of over 99% of all RNA species from bacteria in culture [10]. However, the matter becomes tricky when the bacteria to be analyzed are internalized by host cells, for example, when *S. aureus* invades endothelial cells. Only 1% of total RNA from infected cell is coming from bacteria, and only 5% of that bacterial RNA corresponds to mRNA [11]. Consequently, the likelihood of recovering a representative amount of bacterial mRNA decreases dramatically and leads to skewed results. This aspect is particularly important for a single-cell analysis, which requires minimal contamination with host RNA [11]. In the past, there has not been an emphasis on eliminating endothelial cells including their mRNA prior to bacterial RNA isolation in several protocols. Common procedures involve the lysis of host cells using cold water, acetone:ethanol (1:1), or similar approaches [3,7,12]. Employing a new approach, Raynaud and colleagues have lysed human macrophages and osteoblasts harboring *S. aureus* using PBS 1X/SDS 1% [13]. Lysis was followed by a mechanical disruption of bacterial cells and a chemical-based RNA extraction. Here, we report a protocol adapted from Raynaud and colleagues to recover the mRNA of intracellular *S. aureus* USA300 in a high amount and consistent high quality from human-infected endothelial cells. The advantage of our protocol arises from the fact that we used a manufactured kit followed by the lysis of the endothelial host cells and the mechanical disruption of the cell wall of *S. aureus*. This quick, user-friendly, and straightforward protocol largely avoids the handling of potentially dangerous chemicals and ensures a high reproducibility.

## 2. Material and Methods

The RNA extraction was carried out under two conditions:Condition I: RNA from *S. aureus* in culture after 2 h and 5 h incubationCondition II: RNA extracted from endothelial cells infected with *S. aureus* for 90 min and immediately lysed or further cultured and lysed after 24 h and 48 h, respectively.

### 2.1. Bacteria and Human Endothelial Cell Culture

*Staphylococcus aureus* strain USA300 [14] was grown for 17 h at 37 °C in brain heart infusion broth (Oxoid, Hampshire, UK) with shaking (165 rpm). Bacteria cultures were adjusted to OD = 0.05 (578 nm) and incubated for 2 h or 5 h at 165 rpm and 37 °C, respectively. Afterwards, bacterial cultures were centrifuged at 12,000× *g* for 10 min at 4 °C and washed twice with PBS 1X. Pellets were resuspended in PBS 1X and adjusted to OD = 1 (578 nm). The bacterial cultures used for RNA extraction were centrifuged for 2 min at 12,000× *g* and resuspended in RNA protect^®^ Bacteria Reagent (Qiagen, Hilden, Germany). The samples were incubated for 5 min at room temperature (RT) and centrifuged at 12,000× *g* for 10 min at 4 °C. The supernatant was removed, and the sample was stored at −80 °C until use. For infection assays, bacterial cultures, which were incubated to a midexponential growth phase (2 h) and adjusted to OD = 1 (578 nm), were used.

Human endothelial-like cells EA.hy926 (E-EP-CL-0272.1, Biomol, Hamburg, Germany) were cultured in DMEM (PAN Biotech, Aidenbach, Germany) supplemented with 10% FBS (Bio&Sell, Feucht, Germany) and 100 U/mL penicillin/streptomycin (Pen/Strep; 10,000 units penicillin and 10 mg streptomycin/mL) (Sigma Aldrich, Taufkirchen, Germany). Cells from passage 3 to 18 were used for the experiments.

### 2.2. Bacterial Internalization

EA.hy926 cells were cultured in 25 cm^2^ cell culture flasks (Greiner Bio-One, Frickenhausen, Germany) to a confluence of approximately of 80%. Cells were washed with PBS 1X and an invasion medium was added. The invasion medium consisted of DMEM (PAN Biotech, Aidenbach, Germany) supplemented with 1% human serum albumin (Octapharma, Langenfeld, Germany) and 10 mM HEPES (Sigma Aldrich, Taufkirchen, Germany). Cells were infected with *S. aureus* USA300 grown in BHI to a midexponential growth phase at a multiplicity of infection (MOI) of 100. After 90 min of incubation, cells were washed and a stop medium consisting of DMEM, 10% FBS, and gentamycin (Ratiopharm, Ulm, Germany) with a concentration of 200 µg/mL was added for one hour to eliminate all adherent or extracellular staphylococci. Subsequently, the cells were either lysed after 90 min p.i. (postinfection) or further cultured in fresh culture medium (DMEM containing 10% FBS, and 100 U/mL Pen/Strep) for 24 and 48 h p.i. For these time points, cells were again treated with a stop medium before their lysis. Multiple flasks were pooled to obtain sufficient bacterial RNA for one sample. For 90 min p.i., two flasks were pooled, while for the latter time points at least four flasks were used. To determine the colony forming units (CFUs) of intracellular bacteria per cell, infected cells were washed twice with PBS 1X and lysed with ice-cold H_2_O for 10 min on ice, which was followed by the disruption of the cells using a cell scraper (Sarstedt, Nuembrecht, Germany). The lysate was centrifuged at 4000× *g* for 15 min at 4 °C. Serial dilutions of the lysate were plated onto blood agar plates and incubated overnight at 37 °C. The number of CFUs was determined using a colony counter (colonyQuant, schuett-biotec; Göttingen, Germany). In addition, the number of endothelial cells was assessed for each time point by an automatic cell counter (TC20, Biorad, Feldkirchen, Germany).

### 2.3. RNA Extraction of Endothelial RNA

Endothelial RNA was extracted using the peqGOLD Total RNA Kit (Peqlab; VWR, Darmstadt, Germany) according to the manufacturer’s protocol immediately after each time point.

### 2.4. Extraction of RNA from Intracellular S. aureus

#### 2.4.1. Lysing Endothelial Cells including DNA and RNA

General remarks: Bacteria can quickly adapt their gene expression when conditions change. Therefore, one should work quickly and lyse a maximum of six flasks in parallel. The following protocol refers to 25 cm^2^ cell culture flasks. Upscaling this number of flasks is not recommended since more cell debris will be present, making the extraction more difficult due to a possible attachment of bacteria to the cell debris in later steps.

Remove the cell culture medium and wash with PBS 1X.Add 1 mL 0.02% SDS/PBS 1X and incubate for 20 s (sec). The lysis destroys the endothelial cells including their DNA and RNA.Add 5 mL cold PBS 1X, vigorously flush multiple times, and transfer the sample to a 15 mL falcon tube.Add 1 mL of 0.02% SDS/PBS 1X again and incubate until all cells are detached. The process can be sped up by tilting the flask. At this point, the lysate becomes viscous.Add 7 mL of PBS 1X and vigorously flush multiple times, destroying the endothelial cells and releasing the intracellular bacteria. The use of SDS with PBS 1X avoids the bacterial lysis.Transfer the lysate to the falcon tube.Centrifuge the sample at 4000× *g* for 15 min at 4 °C to pellet the bacteria and further destroy the endothelial cells.Carefully remove the supernatant from the sample and add 1 mL of RNAprotect Bacteria Reagent (Qiagen, Hilden, Germany)Vortex the sample at maximal (max) speed and incubate it for 5 min at RT.Centrifuge at 12,000× *g* for 10 min at RT.Remove the supernatant and store the pellet at −80 °C overnight or up to four weeks.

#### 2.4.2. Mechanical Disruption of *S. aureus* from Culture and Infected Cells

General remark: keep the samples on ice whenever possible to avoid degradation.

Thaw the samples and add 1 mL of RNApro solution (MP Biomedicals, Eschwege, Germany) and vigorously resuspend the pellet until no large clumps are visible.Transfer the mixture to a Lysing Matrix B tube (MP Biomedicals, Eschwege, Germany) and vortex at maximal speed for 10 s.Put the tube in a homogenizer (SpeedMill Plus, Analytikjena, Jena, Germany) and lyse for 3 min twice.Centrifuge the sample for 2 min at 15,000× *g* at 4 °C to pellet the whole bacterial debris.Transfer the supernatant containing bacterial RNA to a new reaction tube.

#### 2.4.3. Purification of *S. aureus* RNA

Bacterial RNA was purified by using the peqGOLD Total RNA Kit, starting with adding one volume of ethanol 70% to the sample and loading the mix onto the RNA column. From this point onwards, the extraction was performed according to the manufacturer’s protocol.

### 2.5. DNAse Treatment

Following RNA extraction, all eukaryotic and prokaryotic samples were treated with DNase using the DNA-free^TM^ Kit (Ambion, Thermofisher Scientific, Harz, Germany). For endothelial RNA, a DNase treatment was carried out according to the manufacturer’s protocol. For bacterial RNA, the protocol was modified. Samples were diluted to a concentration of 200 ng/µL in a reaction volume of a maximum of 0.1 volume of 10X DNase I buffer (not more than 100 µL) in a 0.5 mL tube. Then, 3 µL of recombinant DNase I (rDNase I) was added and the complete sample was incubated for 30 min at 37 °C. Following incubation, a 0.2 volume of DNase Inactivation Reagent were added and centrifuged at 10,000× *g* for 1.5 min. The supernatant containing the RNA was transferred to a new tube. Subsequently, the procedure was repeated. This approach has been tested to be the most effective to eliminate DNA from bacterial samples compared to standard methods where the DNase treatment is recommended to be performed in a column tube. A subsequent cleanup of both endothelial and bacterial samples ensured the removal of any residual ions introduced during the DNase treatment and the concentration of RNA. The cleanup was performed using the RNA Clean & Concentrator^TM^- kit (Zymo Research, Freiburg, Germany). Following DNA depletion, the RNA concentration was measured via a Nanodrop spectrophotometer (ND-1000, Peqlab, VWR, Darmstadt, Germany).

### 2.6. Gel Electrophoresis

The quality of RNA samples was evaluated by gel electrophoresis. A 1% UltraPure^TM^ Agarose (Invitrogen, Thermofisher Scientific, Harz, Germany) gel was prepared using 1X TAE buffer (Thermofisher scientific, Harz, Germany) dissolved in UltraPure^TM^ DEPC treated water (Invitrogen, Thermofisher Scientific, Harz, Germany) and supplemented with 0.5 μg/mL of ethidium bromide (VWR, Darmstadt, Germany). RNA samples and ladder were mixed with an equal volume of RNA loading dye (Thermofisher Scientific, Harz, Germany), incubated at 70 °C for 10 min, and cooled on ice. Subsequently, samples were loaded onto the gel and run for 45 min at 60 V. The eukaryotic 28 S and 18 S rRNAs and prokaryotic 23 S and 16 S rRNA were visualized with a UV transluminator (Gel Doc^TM^ XR, Bio-Rad, Feldkirchen, Germany).

### 2.7. Reverse Transcription (RT)-PCR

Complementary DNA (cDNA) was synthesized using the qScript cDNA Supermix (Quantabio, VWR, Darmstadt, Germany). Each 20 µL of reaction volume contained 1 µg of RNA, 4 µL of qScript cDNA SuperMix (5X), and RNase/DNase-free water, according to the manufacturer’s protocol. Reactions were performed in a T3000 Thermocycler (Biometra) under the following cycling conditions: 25 °C for 5 min, 42 °C for 30 min, and 85 °C for 5 min. RT-PCR products were stored at −20 °C.

### 2.8. Real-Time Quantitative PCR (qPCR)

The cDNA of every sample was diluted to a total amount of ≤100 ng/reaction. cDNA samples were mixed with SYBR Green from the QuantiNova^®^ SYBR^®^ Green PCR Kit (Qiagen, Hilden, Germany) and primers (Metabion, Planegg/Steinkirchen, Germany) using a pipetting robot (Qiagility, Qiagen, Hilden, Germany). Each 25 µL of reaction volume contained 12 µL of cDNA (≤100 ng) and 13 µL of mastermix. The mastermix consisted of 0.25 µL of forward primer, 0.25 µL of reverse primer, and 12.5 µL SYBR Green.

Before the samples were analyzed, all qPCR conditions were optimized; primers were diluted, resulting in C_q_ (quantification cycle or threshold cycle) values between 10 and 20 for reference genes. To ensure the specificity of primers, resulting qPCR fragments were loaded onto an agarose gel and checked for one specific band in the correct size for all investigated genes. All qPCR reactions were performed with a nontemplate control and a no RT-PCR control (only RNA and H_2_O) as a test for DNA contamination. All no RT-PCR reactions yielded C_q_ values ranging from 32 to 39 (depending on the primer pair used). Since a high inter-run variability was often observed in qPCR, we used only one primer pair in one run with all samples. This ensured that housekeeping genes’ C_q_ values could be compared with each other. For the following run using the next primer pair, samples were placed at exactly the same position in the pipetting robot to minimize variability. PCR cycling conditions consisted of 95 °C for 5 min, and 40 cycles of 95 °C for 5 sec and 60 °C for 10 s and were performed in a Rotor-Gene Q cycler (Qiagen, Hilden, Germany). The analysis of the samples was performed using Rotor-Gene Q software (2017, V2.3.1 (Build49), Qiagen, Hilden, Germany).

### 2.9. Data Analysis

The statistical analysis was performed using GraphPad Prism Software (GraphPad Prism version 9.3.1 for Windows, San Diego, CA, USA). The analysis of the upregulation of *ICAM-1* in infected endothelial cells relative to noninfected cells was calculated as follows: first, C_q_ values of every individual sample were normalized with both *GAPDH* and *β-actin* (delta C_q_ = C_q target_ − C_q reference_). Next, the mean of five independent experiments was calculated for each condition. Delta delta C_q_ values were calculated using the formula delta delta C_q_ = delta C_q infected_ − delta C_q non-infected_. Finally, the n-fold change was calculated using the formula 2^-(delta delta C_q_) for each time point [15].

## 3. Results and Discussion

Here, we report a novel strategy for the extraction of *S. aureus* RNA internalized by endothelial cells. First, we extracted RNA from *S. aureus* in culture after a 2 h and 5 h incubation (condition I). Then, endothelial cells were infected with *S. aureus* for 90 min and immediately lysed or further cultured and lysed after 24 h and 48 h, respectively (condition II). Following the internalization of *S. aureus* by the host cells, infected cells were lysed using PBS 1X/SDS 0.02%, based on the protocol established by Raynaud and colleagues [10]. After lysis, we mechanically disrupted the bacterial cell wall and extracted the released RNA with a commercially available kit. Samples were treated with DNase and purified using a cleanup kit. We investigated the stability of housekeeping genes commonly used as reference genes for qPCR with Bestkeeper software (V1, https://www.gene-quantification.de/bestkeeper.html, accessed on 11 March 2023) and calculated the delta C_q_ values for *agr* and *fnbA* in *S. aureus* from culture and intracellular bacteria.

### 3.1. Quality of RNA

As a first step, the quality of the extracted RNA from both host and bacteria was assessed. Figure 1 shows an example of a typical agarose gel image of RNA samples. In this figure, two bands are shown which correspond to the 23 S and 16 S rRNAs extracted from the prokaryotic controls, as well as two bands corresponding to the 28 S and 18 S rRNAs extracted from the eukaryotic control [16,17].
Figure 1Exemplary image of gel electrophoresis of RNA isolated from endothelial cells (host), *S. aureus* from culture and intracellular *S. aureus*. The host lane shows 28 S and 18 S rRNA, while bacterial lanes show 23 S and 16 S rRNA. No host rRNA is detectable in the lane of intracellular *S. aureus*. There were no eukaryotic rRNA bands detected in the lane showing RNA from intracellular *S. aureus*, indicating a low contamination with host RNA. Agarose gels of all individual bacterial samples are shown in Figure 2 and Figure 3. Individual eukaryotic RNA samples are shown in Appendix A and Appendix A. These findings demonstrated the integrity of isolated eukaryotic and bacterial RNA. In addition, the quality was assessed by a Nanodrop measurement, which is frequently used to determine RNA quality (Table 1 and Table 2).
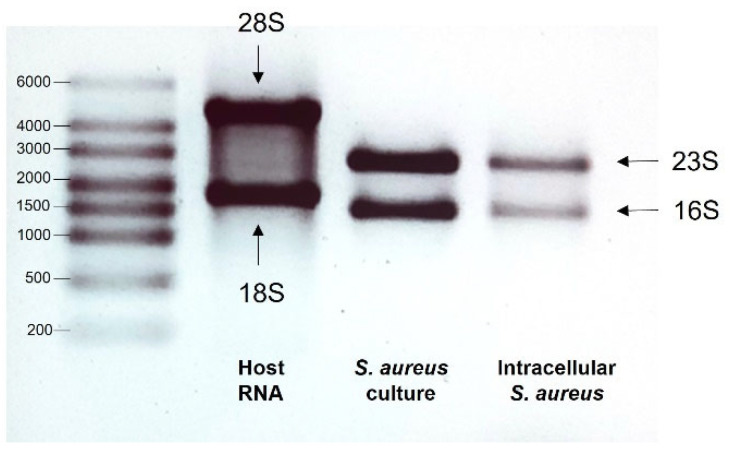

microorganisms-11-01020-t001_Table 1Table 1List of RNA samples of *S. aureus* (shown in Figure 2) in culture after 2 h and 5 h of incubation (condition I) including the amount recovered and the ratios measured via Nanodrop. * Not enough material for gel electrophoresis available.Sample No.SampleAmount (ng/µL)260 nm/280 nm Ratio260 nm/230 nm Ratio1Bacteria from culture 2 h_1234.552.212.292Bacteria from culture 2 h_2255.272.222.473Bacteria from culture 2 h_3371.332.192.524Bacteria from culture 2 h_4411.932.222.525 *Bacteria from culture 5 h_184.072.11.956Bacteria from culture 5 h_290.72.162.37Bacteria from culture 5 h_3129.622.211.078Bacteria from culture 5 h_4126.992.21.779Bacteria from culture 5 h_5123.92.242.29
Figure 2Image of gel electrophoresis of RNA from *S. aureus* from culture after 2 h and 5 h of incubation (condition I) from at least four independent experiments. The 16 S and 23 S (bacterial) rRNA bands are visible. Numbers refer to Table 1’s samples. RNA ladder in bp. The amount detected on gel does not correspond with the amount measured via Nanodrop. The sample 5 has not enough material and it is not shown.
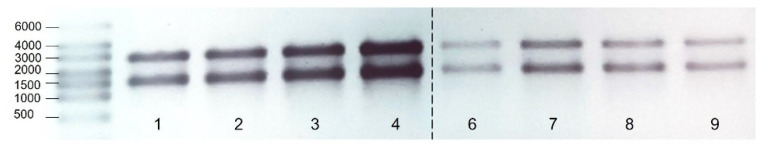

microorganisms-11-01020-t002_Table 2Table 2List of RNA samples of intracellular *S. aureus* (gel shown in Figure 3) 90 min, 24 h, and 48 h postinfection (condition II) including the amount recovered and the ratios measured via Nanodrop. * Not enough material for gel electrophoresis was available.Sample No.SampleAmount (ng/µL)260 nm/280 nm Ratio260 nm/230 nm Ratio1Intracellular bacteria 90 min p.i._1112.012.313.142Intracellular bacteria 90 min p.i._2137.132.332.663Intracellular bacteria 90 min p.i._3136.162.32.214Intracellular bacteria 90 min p.i._4197.182.122.185 *Intracellular bacteria 90 min p.i._5233.042.251.426 *Intracellular bacteria 24 h p.i._179.232.332.717 *Intracellular bacteria 24 h p.i._285.822.262.878Intracellular bacteria 24 h p.i._3250.782.222.549Intracellular bacteria 24 h p.i._4195.492.242.5510Intracellular bacteria 24 h p.i._5240.662.192.7811Intracellular bacteria 48 h p.i._1212.662.182.4212 *Intracellular bacteria 48 h p.i._288.212.270.413Intracellular bacteria 48 h p.i._3169.42.292.9914Intracellular bacteria 48 h p.i._4182.992.20.8815Intracellular bacteria 48 h p.i._5138.572.292.66
Figure 3Image of gel electrophoresis of RNA samples from intracellular *S. aureus* 90 min, 24 h, and 48 h postinfection (condition II) from five independent experiments. The 16 S and 23 S (bacterial) rRNA bands are visible. Numbers refer to Table 2’s samples. RNA ladder in bp. The amount detected on gel does not correspond with the amount measured via Nanodrop.
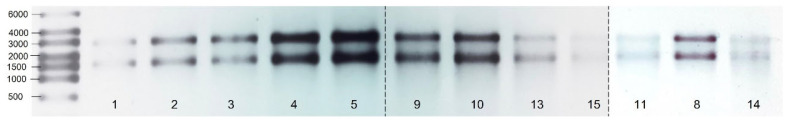



The ratio of absorbance at 260 nm and 280 nm, and the ratio of absorbance at 260 nm and 230, respectively, should give information about the purity of RNA. According to the Nanodrop manufacturer, acceptable 260/280 ratios should range between 1.8 and 2.0, and 260/230 ratios should range between 2.0 and 2.2, respectively. After the cleanup step, all bacterial samples showed 260/280 ratios exceeding 2.0, which could indicate contamination with, for example, protein and guanidine. The 260/230 ratios ranged from 0.4 up to 2.99, which indicated a contamination with carbohydrates and phenol among other substances. Since the contamination of RNA samples can lead to the inhibition of the subsequent cDNA synthesis, we performed a two-step RT-qPCR to analyze the variability of the samples.

### 3.2. Stability of Housekeeping Genes

Next, we evaluated the stability of housekeeping genes to be used as reference genes in SYBR Green qPCR assays. It is generally recommended to use more than one reference gene to increase the reliability of generated results, since a housekeeping gene may be affected under any given experimental condition [18,19,20]. Any variation in the reference gene can potentially influence the gene expression [21]. Housekeeping genes might be differentially regulated when bacteria are internalized by host cells compared to bacteria grown in culture. There is no real consensus on which *S. aureus* housekeeping genes are suitable to be used as reference genes for the investigation of the intracellular life style of *S. aureus*. However, *gyrB*, *aroE*, *tmRNA*, *gmk*, and *hu* have been frequently used for gene expression analysis in *S. aureus* [7,13,22,23]. Therefore, we analyzed the stability of these genes in bacterial cultures after 2 h and 5 h of incubation, and in intracellular bacteria recovered from infected endothelial cells after 90 min, 24 h, and 48 h postinfection. The housekeeping genes are listed in Table 3 and Appendix A and were ranked according to their stability calculated by Bestkeeper software (Table 4).

As a measurement of stability, the standard deviation (std dev) (±C_q_) was used. The stability of housekeeping genes was determined for condition I and II separately, and across both conditions. Figure 4A visualizes the C_q_ values of all five selected housekeeping genes for *S. aureus* from culture after 2 h and 5 h of incubation (condition I). A std dev (± C_q_) exceeding 1 was considered too large for a gene to be used as reference gene. According to *Bestkeeper, gmk* was the most stable gene across both time points with a std dev (± C_q_) of 0.32, while *aroE* was the least stable with a std dev (±C_q_) of 1.32 (Table 4). Therefore, all housekeeping genes, except of *aroE,* qualified as reference genes to analyze the expression of bacteria from culture. In a similar fashion, Figure 4B shows the C_q_ values of the housekeeping genes for intracellular *S. aureus* across the three time points after internalization (condition II). In contrast to bacteria from culture, the most stable gene was *gyrB* with a std dev (± C_q_) of 0.26, while the std dev (± C_q_) obtained for *gmk* and *aroE* were variable with a value greater than one (Table 4). Subsequently, we evaluated the stability of genes across both conditions I and II (Figure 4C and Table 4). The genes *gyrB* and *tmRNA* appeared as the most stable genes, while *aroE* was the least stable. To find genes that are stable across many conditions is especially important when gene expression before and after the internalization of *S. aureus* by host cells is compared. Of note, other housekeeping genes not tested in this analysis might be suitable as reference genes as well. In addition, we performed a small analysis and only used Bestkeeper, which is free of charge. Other software are available, for example GeNorm or Normfinder, which are also frequently used. It has been reported that those software tools analyze the stability of genes differently, resulting in a different ranking of genes [23].

### 3.3. Normalization of Target Genes

After determining the appropriate reference genes for our system, we calculated delta C_q_ values to normalize the data. According to our experience, some of the genes most influenced during growth in culture and intracellular survival of *S. aureus* are those related to the quorum sensing operon called accessory gene regulator (Agr) [24,25]. Agr is an operon composed of four genes *agrA, B, C,* and *D* and is responsible for the activation and downregulation of several genes [26]. Among the genes downregulated by Agr are the adhesins of *S. aureus* including fibronectin A as the most representative one [27,28]. Furthermore, we found that these genes were affected depending on the different steps of infection [24]. Thus, these two genes were selected to normalize our data. When analyzing these data, some aspects regarding RNA degradation needed consideration. As shown in Table 5 and Table 6, the std devs (±C_q_) of the target genes *agrA* and *fnbA* were within an acceptable range for both bacteria from culture and intracellular bacteria. Thus, low std devs of reference and target genes were found.

The selection of reference genes is not the unique factor that introduces a variation of delta C_q_ values. Partly degraded RNA can nevertheless lead to a large variation in delta C_q_ values if, for example, by chance only, the mRNA of a target gene was affected but not the mRNA of the reference gene. Therefore, a low std dev of the delta C_q_ values can provide information about the quality of the extracted RNA. Of note, it was reported that RNA degradation rates were relatively equal across large transcripts and, therefore, did not have a substantial impact on the gene expression analysis [25]. In that study, underrepresented mRNA transcripts had an average transcript length of 2000 bp and were significantly shorter than normally expressed genes. This result led to the assumption that larger transcripts were more stable and less prone to degradation. The ranking according to the length of the housekeeping genes tested in our study is shown in Appendix A. All housekeeping transcripts (whole gene) were smaller than 2000 bp and there was no clear correlation between the transcript length and the obtained ranking according to the std dev. There are two possible explanations for this observation: (i) housekeeping genes are differentially regulated in *S. aureus* from culture and intracellular *S. aureus* and at different time points within those conditions, or (ii) RNA degradation rates are not equal across small bacterial transcripts and conditions. Moreover, a combination of both aspects is conceivable. Regardless, RNA degradation may have a major impact in producing variability in qPCR, which underlines the importance of good bacterial RNA quality. Target genes were normalized by subtracting the C_q_ value of a reference gene from the corresponding C_q_ value of the target gene of the same sample (delta C_q_ = C_q_ target − C_q_ reference). The calculation was performed for each reference gene individually. The resulting delta C_q_ values for bacteria from culture are shown in Figure 5A (*agrA*), Figure 5B (*fnbA*), and Table 7. The lowest calculated std dev was 0.33 and the highest 1.47. Results from intracellular bacteria are shown in Figure 6A (*agrA*), Figure 6B (*fnbA*), and Table 8. The lowest calculated std dev was 0.18 and the highest std dev detected was 1.26. Taken together, the variability of delta C_q_ for both *S. aureus* from culture and intracellular *S. aureus* were relatively low and suggested a high RNA quality and low RNA degradation.

To check that host cells responded to the infection, we investigated the adhesion molecule ICAM-1, which is well known for driving immune responses to inflammation [29]. The normalized data for ICAM-1 are shown in Figure 7. ICAM-1 was normalized with the commonly used reference genes GADPH (Figure 7A) and β-actin (Figure 7B), which showed a highly similar distribution of values [30,31]. We applied the delta delta C_q_ method to calculate the relative change in gene expression between noninfected and infected cells, whereby both reference genes were used (Figure 8). The expression of ICAM-1 was only slightly increased at 90 min p.i.; however, it was upregulated 20-fold at 24 h and 48 h p.i., which is consistent with previous findings [24].

### 3.4. Applicability and Limitations of This Protocol

In this study, we extracted the RNA of intracellular *S. aureus* 90 min, 24 h, and 48 h after infection. The number of bacteria recovered from host cells determines the amount of extracted RNA. We assessed the amount of *S. aureus* internalized by endothelial cells at 90 mi, 24 h, and 48 h p.i. (Figure 9). We found a significant decrease of intracellular bacteria, dropping from about 11 CFUs/cell to 4 CFUs/cell at 24 h p.i. and 1 CFU/cell after 48 h p.i. (mean values). It is known that the number of intracellular bacteria further decreases during the following five days of infection [32]. This has important implications for our protocol. While it can be used for intracellular *S. aureus* for up to 48 h p.i., it may not be suitable for later time points. Less bacteria result in less RNA recovery and, therefore, requires an increased number of host cells. Scaling up to bigger flasks also increases host cell debris, which can inhibit a sufficient breakdown of the bacterial cell wall and access to bacterial RNA during mechanical disruption. In addition, the current method is not suitable for a large number of samples. In this study, RNA was isolated simultaneously from a maximum of six 25 cm^2^ cell culture flasks. Isolating RNA from a larger number of flasks leads to more time needed for each step and promotes RNA degradation. Although the Nanodrop values of most of the bacterial samples exceeded the acceptable range, the variability of the delta C_q_ values was low. Therefore, we conclude that these values should be treated with caution as also proposed by other groups [33]. The question must be whether the components that absorb at 280 and 230 nm have an impact on the downstream application. In this study, there was no significant interference detected by RT-qPCR. In our opinion, the decision on whether RNA has a sufficient quality for qPCR should be based on the appearance of rRNA in the agarose gel image and other analysis methods such as capillary-electrophoresis systems [34]. The RNA isolation protocol presented in this paper must be adapted for every cell type individually. The lysis buffer must be strong enough to lyse host cells but gentle enough to leave bacterial cells intact. Thus, small changes related to the time of exposition to the lysing buffer might be necessary to extend this protocol to other type of cells.

## 4. Conclusions

Here, we presented a novel strategy for the isolation of RNA from *S. aureus* internalized by endothelial cells. The extraction protocol adapted from Raynaud and colleagues allowed an efficient and high-quality extraction of RNA from intracellular *S. aureus* during the early phase of infection. We identified appropriate housekeeping genes as reference genes and provided normalized qPCR data with a relatively low variability. Lysis with SDS allowed a sufficient removal of excessive human RNA and DNA and subsequently the specific capture of bacterial RNA using a commercially available column-based kit. In our protocol, the SDS concentration was reduced to 0.02%. The extracted RNA proved to be of good quality according to a gel electrophoresis analysis. Using quantitative real-time PCR (qPCR), we verified the reproducibility of our protocol. The *S. aureus* genes *agrA* and *fnbA* were investigated, which play major roles during acute and chronic infection and the invasion of host cells [24]. Housekeeping genes suitable as reference genes in terms of stability were identified using Bestkeeper software [18]. However, this program did not represent an in-depth investigation. Further studies are needed to include other reference genes and use multiple software tools to rank them according to their stability. Nevertheless, the established protocol enabled the generation of reliable and reproducible gene expression data and facilitated the study of the host–pathogen relationship. The generated qPCR data demonstrated a low variability of delta C_q_ values of *agrA* and *fnbA* in intracellular *S. aureus*. As a control, the RNA of host cells was extracted and analyzed for the expression of *ICAM-1* upon infection. We conducted our qPCR experiments according to the MIQE guidelines whenever applicable [35].

Since the protocol is less useful for later stages of infection, there is still a need for the development of techniques that are more sensitive and more efficient when enriching small amounts of bacterial RNA. Taken together, our newly established protocol can be used to generate meaningful qPCR data for the gene expression analysis of intracellular *S. aureus* and potentially other Gram-positive bacteria. Unravelling the interplay of human host cells and intracellular *S. aureus* can pave the way to find new treatment strategies for acute and chronic infections and thus to get an upper hand on this increasing threat to human health.

## Figures and Tables

**Figure 4 microorganisms-11-01020-f004:**
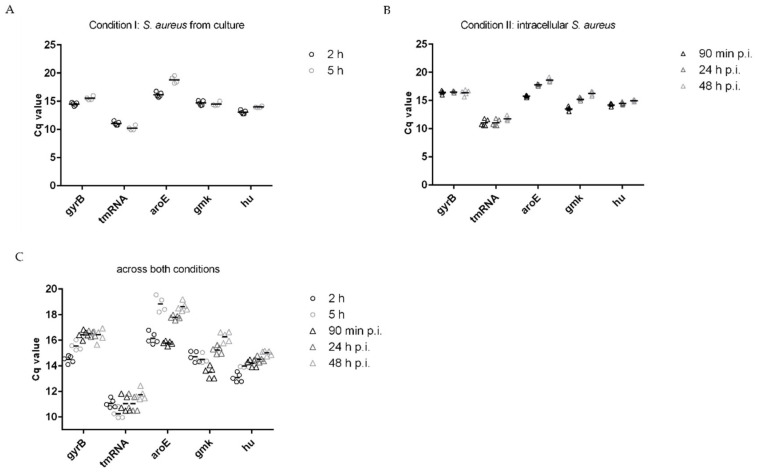
Variability of C_q_ values of candidate housekeeping genes for (**A**) *S. aureus* from culture after 2 h and 5 h of incubation (condition I), (**B**) intracellular *S. aureus* at 90 min, 24 h, and 48 h postinfection (condition II), and (**C**) across conditions I and II. Bars indicate the mean of at least four independent experiments.

**Figure 5 microorganisms-11-01020-f005:**
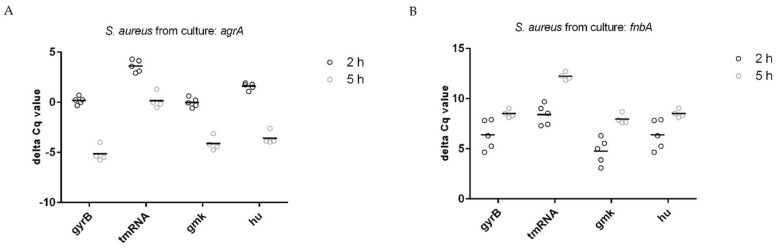
Variability of delta C_q_ values of *S. aureus* from culture 2 h and 5 h after incubation (condition I) for each reference gene individually for (**A**) *agrA* and (**B**) *fnbA*. Bars indicate the mean of at least four independent experiments.

**Figure 6 microorganisms-11-01020-f006:**
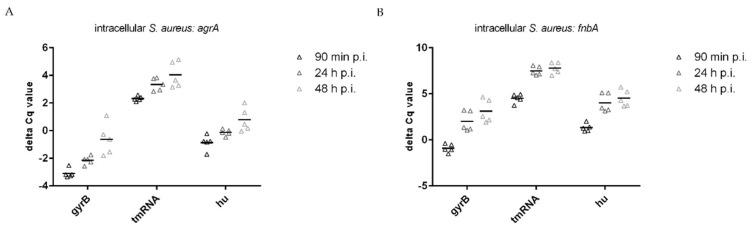
Variability of delta C_q_ values of intracellular *S. aureus* at 90 min, 24 h, and 48 h postinfection (condition II) for each reference gene individually for (**A**) *agrA* and (**B**) *fnbA*. Bars indicate the mean of five independent experiments.

**Figure 7 microorganisms-11-01020-f007:**
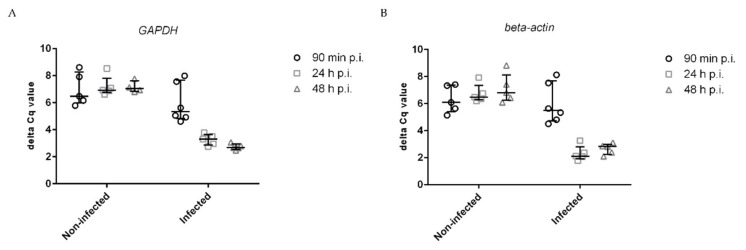
Delta C_q_ values of *ICAM1* in noninfected and infected endothelial cells. Normalized data using (**A**) *GADPH* and (**B**) *β-actin*. Bars indicate the std dev of five independent experiments.

**Figure 8 microorganisms-11-01020-f008:**
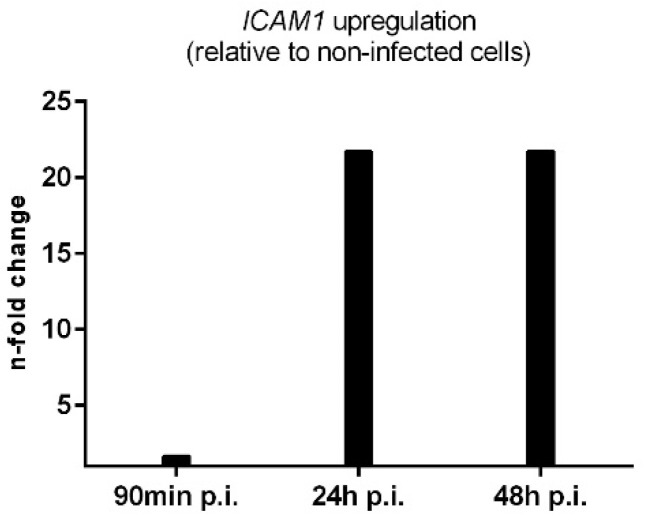
Change of gene expression of *ICAM-1* in *S. aureus* infected endothelial cells relative to noninfected cells.

**Figure 9 microorganisms-11-01020-f009:**
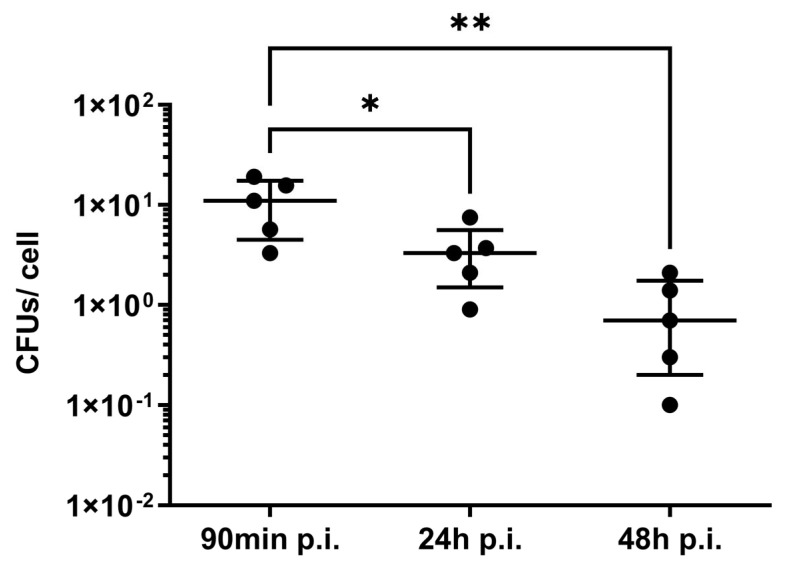
CFUs per cell of intracellular *S. aureus* recovered after 90 min, 24 h, and 48 h postinfection. One-way ANOVA with Tukey’s multiple comparisons test; * *p* < 0.05, ** *p* < 0.01.

**Table 3 microorganisms-11-01020-t003:** Overview of investigated bacterial housekeeping genes and target genes.

Gene	Gene Product	Locus Tag
**Housekeeping genes**		
*gyrB*	DNA gyrase subunit B	SAUSA300_RS00030
*tmRNA*	tmRNA	SAUSA300_RS15205
*aroE*	Shikimate dehydrogenase (NADP(+))	SAUSA300_RS08475
*gmk*	Guanylate kinase	SAUSA300_RS05970
*hu*	Nucleotide-associated protein HU	SAUSA300_RS07430
**Target genes**		
*agrA*	Quorum-sensing response regulator AgrA	SAUSA300_RS10950
*fnbA*	Fibronectin-binding protein A	SAUSA300_RS13530

**Table 4 microorganisms-11-01020-t004:** Ranking of bacterial housekeeping genes. n: number of independent experiments carried out for each condition; std dev: standard deviation; C_q_: crossing point. Values highlighted in grey were excluded from further analysis.

Ranking	Housekeeping Gene	n	std dev (±C_q_)
** Condition I: *S. aureus* from culture **			
1	*gmk*	9	0.32
2	*tmRNA*	9	0.43
3	*hu*	9	0.46
4	*gyrB*	9	0.51
5	*aroE*	9	1.32
** Condition II: intracellular *S. aureus* **			
1	*gyrB*	15	0.26
2	*hu*	15	0.32
3	*tmRNA*	15	0.55
4	*gmk*	15	1.02
5	*aroE*	15	1.08
** Across both conditions (I + II) **			
1	*gyrB*	24	0.56
1	*tmRNA*	24	0.56
2	*hu*	24	0.73
3	*gmk*	24	0.81
4	*aroE*	24	1.18

**Table 5 microorganisms-11-01020-t005:** Standard deviation of Cq values of target genes of *S. aureus* from culture after 2 h and 5 h of incubation (condition I). n: number of independent experiments carried out for each condition; std dev: standard deviation; C_q_: crossing point.

Target Gene	n	std dev (± C_q_)
**2 h**		
*agrA*	5	0.25
*fnbA*	5	0.93
**5 h**		
*agrA*	4	0.43
*fnbA*	4	0.35

**Table 6 microorganisms-11-01020-t006:** Std dev of target genes of intracellular *S. aureus* at 90 min, 24 h, and 48 h postinfection (condition II). n: number of independent experiments carried out for each condition; std dev: standard deviation; C_q_: crossing point.

Target Gene	n	std dev (± C_q_)		
		**90 min p.i.**	**24 h p.i.**	**48 h p.i.**
*agrA*	5	0.42	0.18	0.64
*fnbA*	5	0.38	0.95	0.70

**Table 7 microorganisms-11-01020-t007:** Mean and std dev (in brackets) of normalized target gene expression for individual housekeeping genes in *S. aureus* from culture after 2 h and 5 h of incubation (condition I).

Reference Gene	Target Gene: *agrA*	
	**2 h**	**5 h**
*gmk*	−0.01 (0.46)	−4.125 (0.7)
*tmRNA*	3.62 (0.61)	0.15 (0.8)
*hu*	1.61 (0.33)	−3.58 (0.64)
*gyrB*	0.18 (0.38)	−5.15 (0.79)
	**Target gene: *fnbA***	
*gmk*	4.77 (1.29)	7.96 (0.51)
*tmRNA*	8.4 (1.02)	12.24 (0.37)
*hu*	6.39 (1.47)	8.51 (0.39)
*gyrB*	4.96 (0.95)	6.94 (0.33)

**Table 8 microorganisms-11-01020-t008:** Mean and std dev (in brackets) of normalized target gene expression for individual housekeeping genes in intracellular *S. aureus* at 90 min, 24 h, and 48 h postinfection (condition II).

Reference Genes	Target Gene: *agrA*		
	90 min p.i.	24 h p.i.	48 h p.i.
*tmRNA*	2.32 (0.18)	3.35 (0.45)	4.04 (0.94)
*hu*	−0.85 (0.53)	−0.13 (0.22)	0.79 (0.86)
*gyrB*	−3.09 (0.33)	−2.14 (0.3)	−0.63 (1.15)
	**Target gene: *fnbA***		
	**90 min p.i.**	**24 h p.i.**	**48 h p.i.**
*tmRNA*	4.51 (0.48)	7.49 (0.48)	7.79 (0.61)
*hu*	1.33 (0.43)	4.02 (1.01)	4.53 (0.91)
*gyrB*	0.9 (0.46)	2.0 (1.1)	3.12 (1.26)

## Data Availability

Not applicable.

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
