# Peer review of "Extraction of High-Quality RNA from S. aureus Internalized by Endothelial Cells"

_microorganisms, 2023, doi:10.3390/microorganisms11041020_

Round 1

Reviewer 1 Report

The authors report on the adaptation and advanced development of a method to extract high-quality RNA from bacterial cells (S. aureus), which have been internalized by host cells. This robust and useful protocol fills a gap which has been a major hurdle until now for transcriptomic investigations on intracellular pathogens.

Major comments

1.       Material and Methods: An exact description of the conditions I and II should be given in this section.

2.       Internalization of S. aureus cells and switch to an intracellular lifestyle can lead to the formation of specific phenotypes, esp. the small colony-phenotype known for chronic and relapsing infections. This should be mentioned in the Introduction and/or Discussion section to emphasize the particular value of the method presented here for investigations of this phenotype.

3.       The authors used the endothelial-like cell line EA.hy926 as host cell. Could the authors speculate whether the protocol would work for other cell types, esp. epithelial cells?

Minor comments

4.       Latin designations of the bacteria should be italicized throughout the text.

5.       L97: gentamicin (sic!)

6.       L100: Pen/Strep, please explain abbreviation

Author Response

Thank you very much for your consideration of our manuscript and request for a revised version. We believe that all raised concerns in our initial submission were sound and may improve the quality of the manuscript.

To avoid confusions, we submitted as well a new version with accepted corrections (please see the attachment). We apologise that the previous version submitted had several formatting errors.

We have now addressed all the questions raised by the referees. We would also like to thank the referees for their thorough evaluation of our work and for their helpful comments and constructive criticism. All the changes are clearly specified and highlighted in the revised version of the manuscript to facilitate their identification.

We hope we have answered adequately to all the queries and look forward your response.

Sincerely,

Lorena Tuchscherr

Reviewer 1:

The authors report on the adaptation and advanced development of a method to extract high-quality RNA from bacterial cells (S. aureus), which have been internalized by host cells. This robust and useful protocol fills a gap which has been a major hurdle until now for transcriptomic investigations on intracellular pathogens.

We would like to thank the reviewer for spending time to correct our manuscript. We apologize that the previous version submitted had several formatting errors.

Major comments

  1. Material and Methods: An exact description of the conditions I and II should be given in this section.

We would like to thank the reviewer for this point. This information was added at the beginning of the section Mat. and Methods.

  1. Internalization of S. aureus cells and switch to an intracellular lifestyle can lead to the formation of specific phenotypes, esp. the small colony-phenotype known for chronic and relapsing infections. This should be mentioned in the Introduction and/or Discussion section to emphasize the particular value of the method presented here for investigations of this phenotype.

We agree with the reviewer and this information was included in our introduction:

Within host cells, S. aureus is able to switch to a dormant bacterial phenotype called small colony variants (SCVs). This bacterial phenotype is less sensitive to antimicrobial treatment and related to S. aureus persistence

  1. The authors used the endothelial-like cell line EA.hy926 as host cell. Could the authors speculate whether the protocol would work for other cell types, esp. epithelial cells?

Yes, this speculation was included in the last paragraph:

Thus, small changes related to time of exposition to the lysing buffer might be necessary to extend this protocol to other type of cells.

Minor comments

  1. Latin designations of the bacteria should be italicized throughout the text.
  2. L97: gentamicin (sic!)
  3. L100: Pen/Strep, please explain abbreviation

Thank you for these suggestions. The mistakes were modified.

Reviewer 2 Report

This manuscript outlines the RNA extraction method for S. aureus internalized by endothelial cells and presents data on RNA concentration and quantity. The authors also evaluated various housekeeping genes in S. aureus by examining the standard deviation of Cq values and ultimately excluded the aroE gene. The method described in this study has the potential to enhance the quality of RNA extraction from internalized S. aureus.

Parts of the text appear disjointed or unclear. Kindly revise the writing to ensure a smoother flow and improved coherence. (Examples: line 153-154, line 158, line 386-387, line 411)

Please check the comments below:

Please check the words Staphylococcus, S. aureus to be italicized.

Please check the gene names (gmk, tmRNA, aroE, gmk, hu, agrA, fnbA) to be italicized.

Please review the organization of the text, figures, and table footnotes

Line 63-74: Please revise this section to serve as the introduction. Consider relocating some statements to the discussion portion and double-check for any redundancies in the writing.

Line 83: Quiagen -> Qiagen

Line 100: Abbreviations should be explained in full at their first appearance. (Pen/Strep)

 Also, provide the concentration of the antimicrobials you used (Final 1% volume using 10,000 U/mL solution?).

Line 106: Please add a speed (xg) for the centrifugation

Line 115-145: (suggestion) 2.4.1. and 2.4.2. parts can be more modified as a flow chart or table listing the procedure with numbers for better readability.

Line 159: Compared to what kinds of other settings, this method was proven to be ‘the most effective’?

Line 182-185: If the total ‘amount’ of cDNA was 100ng in 25uL reaction, please revise the sentence.

Line 207: Please maintain a consistent tense (past) throughout the sentences in the text. (is -> was)

Line 223: a typical agarose gel. -> a typical agarose gel image of RNA samples.

Line 224-229: Please adjust the text and image (Figure 1) (Applying ~ to).

Line 288: What was the threshold and how did you choose that?

Line 304-311: Please delete the table duplicated (Table 5).

Table 4, 5: ‘std dev [+- Cq]’ may hard to understand for readers. Please explain it at the footnote.

Table 5: Std dev should be explained in full (Standard deviation of Cq values) for the title of the table.

Line 319: The (italic) -> The

Figure 9: Statistical method and program should be explained in the materials and methods part.

Line 411-412: Please connect two sentences into one for better reading.

Author Response

Thank you very much for your consideration of our manuscript and request for a revised version. We believe that all raised concerns in our initial submission were sound and may improve the quality of the manuscript.

To avoid confusions, we submitted as well a new version with accepted corrections (please see the attachment). We apologize that the previous version submitted had several formatting errors.

We have now addressed all the questions raised by the referees. We would also like to thank the referees for their thorough evaluation of our work and for their helpful comments and constructive criticism. All the changes are clearly specified and highlighted in the revised version of the manuscript to facilitate their identification.

We hope we have answered adequately to all the queries and look forward your response.

Sincerely,

Lorena Tuchscherr

Reviewer 2:

This manuscript outlines the RNA extraction method for S. aureus internalized by endothelial cells and presents data on RNA concentration and quantity. The authors also evaluated various housekeeping genes in S. aureus by examining the standard deviation of Cq values and ultimately excluded the aroE gene. The method described in this study has the potential to enhance the quality of RNA extraction from internalized S. aureus.

We would like to thank the reviewer for her/his time to correct our manuscript

Parts of the text appear disjointed or unclear. Kindly revise the writing to ensure a smoother flow and improved coherence. (Examples: line 153-154, line 158, line 386-387, line 411)

We would like to thank the reviewer for this remark. The sentences were reformulated

Please check the comments below:

Please check the words StaphylococcusS. aureus to be italicized.

Please check the gene names (gmk, tmRNA, aroE, gmk, hu, agrA, fnbA) to be italicized.

Please review the organization of the text, figures, and table footnotes

 All the suggestions/comments were corrected and included in the manuscript

Line 63-74: Please revise this section to serve as the introduction. Consider relocating some statements to the discussion portion and double-check for any redundancies in the writing.

Line 83: Quiagen -> Qiagen

Line 100: Abbreviations should be explained in full at their first appearance. (Pen/Strep)

 Also, provide the concentration of the antimicrobials you used (Final 1% volume using 10,000 U/mL solution?).

Line 106: Please add a speed (xg) for the centrifugation

Line 115-145: (suggestion) 2.4.1. and 2.4.2. parts can be more modified as a flow chart or table listing the procedure with numbers for better readability.

We appreciate the suggestions of the reviewer. All the mistakes were corrected.

Line 159: Compared to what kinds of other settings, this method was proven to be ‘the most effective’?

We agree with the reviewer and the sentence was completed:

This approach has been tested to be the most effective to eliminate DNA from bacterial samples compared to standard methods where the DNAase treatment is recommended to be performed into a column-tube.

Line 182-185: If the total ‘amount’ of cDNA was 100ng in 25uL reaction, please revise the sentence.

We thank the reviewer for this comment. The sentence was modified:

cDNA of every sample was diluted to a total amount of ≤100 ng/reaction. cDNA samples were mixed with SYBR Green from the QuantiNova® SYBR® Green PCR Kit (Qiagen) and primers (Metabion) using a pipetting robot (Qiagility, Qiagen). Each 25 µl reaction volume contained 12 µl of cDNA (≤100 ng) and 13 µl of mastermix. The mastermix consisted of 0.25 µl of forward primer, 0.25 µl of reverse primer and 12.5 µl SYBR Green

Line 207: Please maintain a consistent tense (past) throughout the sentences in the text. (is -> was)

Line 223: a typical agarose gel. -> a typical agarose gel image of RNA samples.

Line 224-229: Please adjust the text and image (Figure 1) (Applying ~ to).

All these points have been corrected

Line 288: What was the threshold and how did you choose that?

We apologize for this error. The sentence has been modified:

In contrast to bacteria from culture, the most stable gene was gyrB with a std dev [± Cq] of 0.26, while the Cq obtained for gmk and aroE were variable with a std dev greater than one.

Line 304-311: Please delete the table duplicated (Table 5).

Table 4, 5: ‘std dev [+- Cq]’ may hard to understand for readers. Please explain it at the footnote.

Table 5: Std dev should be explained in full (Standard deviation of Cq values) for the title of the table.

Line 319: The (italic) -> The

Thank you for notifying us of these errors. All of them have been corrected

Figure 9: Statistical method and program should be explained in the materials and methods part.

We agree with the reviewer. This information is included as a point 2.9 in the materials and methods.

Line 411-412: Please connect two sentences into one for better reading.

Thank you for your suggestion. The sentences have been corrected ((please see the file of the manuscript enclosed).

Reviewer 3 Report

In this manuscript, authors reported a novel protocol for extraction of high-quality RNA from Staphylococcus aureus internalized by endothelial cells.

The manuscript is well described, and materials and methods applications are appropriate.

This reviewer noticed some minor points  as follows:

1.      There are other housekeeping genes and target genes for S. aureus. Why did authors choose gyrB, tmRNA, aroE, gmk, hu (housekeeping genes) and agrA, fnbA (target genes)? It should be clarified in the text.

2.      Table 1 and Table 2, Sample Nr should be read as Sample No.

3.      Table 3, subheading of house keeping genes and target genes should be moved to first column.

4.      Bacteria name and all the genes’ names should be italicized. Please check them throughout the manuscript.

5.      Line 304-311, table and text are mixed and please check the accuracy of the content.

6.      What are the limitations of this study?

Author Response

Thank you very much for your consideration of our manuscript and request for a revised version. We believe that all raised concerns in our initial submission were sound and may improve the quality of the manuscript.

To avoid confusions, we submitted as well a new version with accepted corrections (please see the attachment). We apologize that the previous version submitted had several formatting errors.

We have now addressed all the questions raised by the referees. We would also like to thank the referees for their thorough evaluation of our work and for their helpful comments and constructive criticism. All the changes are clearly specified and highlighted in the revised version of the manuscript to facilitate their identification.

We hope we have answered adequately to all the queries and look forward your response.

Sincerely,

Lorena Tuchscherr

Round 2

Reviewer 1 Report

All comments have been addressed.

Author Response

Thank you for your time. 

Reviewer 2 Report

Although the authors have made revisions to the manuscript as suggested, it remains somewhat challenging to identify the specific changes made. To address this issue, please provide a point-by-point response to the comments. Additionally, ensure that the formatting of the references adheres to the guidelines established by this journal.

Author Response

Dear reviewer, 

we attached in the last revision our comments. Please find the point-by-point response for your comments attached. The references were modify to the ACS style 

Best regards

Lorena Tuchscherr

Reviewer 3 Report

Thank you for the revised manuscript.

Authors modified the manuscript as per reviewers' recommendation.

No more comments from this reviewer.

Author Response

Thank you for your time